# PERTURBATION GUIDED DRUG MOLECULE DESIGN VIA LATENT RECTIFIED FLOW

## ABSTRACT

Phenotypic drug discovery generates rich multi-modal biological data, yet translating complex cellular responses into molecular design remains a computational bottleneck. Existing generative methods operate on single modalities (transcriptomic or morphological alone) and condition on post-treatment measurements without leveraging paired control-treatment dynamics. We present **Pert2Mol**, the first framework for multi-modal phenotype-to-structure generation that integrates transcriptomic and morphological features from paired control-treatment experiments. Pert2Mol employs separate ResNet and cross-attention encoders for microscopy images and gene expression profiles, with bidirectional cross-attention between control and treatment states to capture perturbation dynamics rather than simple differential measurements. These multi-modal embeddings condition a rectified flow transformer that learns velocity fields along straight-line trajectories from noise to molecular structures, enabling deterministic generation with superior efficiency over diffusion models. We introduce Student-Teacher Self-Representation (SERE) learning where an exponential moving average teacher supervises student representations across network depths, stabilizing training in high-dimensional multi-modal spaces. Unlike previous approaches that require preprocessed differential expression vectors, Pert2Mol learns perturbation effects directly from raw paired experimental data. Experiments on large-scale datasets demonstrate the first successful multi-modal framework for phenotype-driven molecular generation.

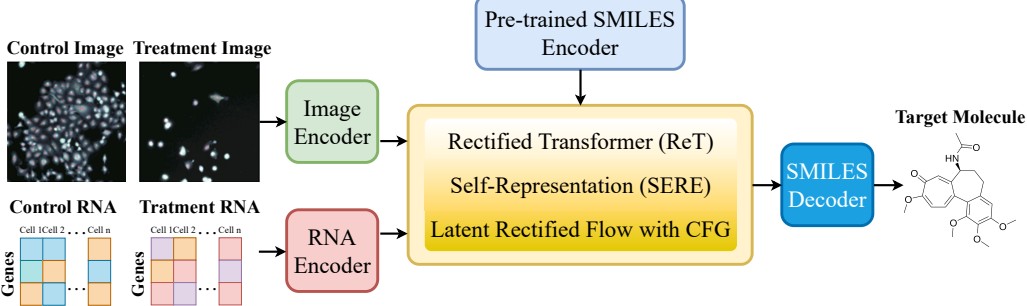

Figure 1: Perturbation guided drug molecule design through Pert2Mol

## 1 INTRODUCTION

Phenotypic drug discovery is re-emerging as a powerful alternative to target-centric strategies, consistently yielding more first-in-class medicines Swinney (2013); Vincent et al. (2022); Moffat et al. (2017). Modern assays now capture transcriptomic and morphological responses to perturbations Chandrasekaran et al. (2021); Bray et al. (2017); Haghighi et al. (2022); Way et al. (2021), yet translating these rich readouts into molecular design remains a challenging inverse problem. Current workflows rely on manual interpretation, creating a bottleneck between phenotypic data and structure–activity insights. This mapping is inherently many-to-many: distinct compounds often induce

convergent phenotypes Sun et al. (2012); Schneider et al. (1999). Rather than a limitation, this redundancy enables scaffold hopping and novel mechanism discovery, reflecting how most approved drugs historically emerged from phenotype-first approaches Eder et al. (2014).

Existing generative methods cannot capture this complexity. Transcriptome-only models such as MolGene-E use differential expression profiles without paired controls Ohlan et al. (2025), while morphology-based approaches rely solely on imaging Zapata et al. (2023); Tang et al. (2025); Caicedo et al. (2017). No framework integrates both modalities, despite their complementarity: expression reveals pathway-level changes, while microscopy captures structural phenotypes absent from gene-level data Scheeder et al. (2018); Rohban et al. (2017). Graph-based Simonovsky & Komodakis (2018); Mitton et al. (2021) and text-conditioned models Chang & Ye (2024); Wang et al. (2023); Edwards et al. (2022) ensure structural validity or semantic alignment but fail on high-dimensional multi-modal perturbation data. Multi-modal conditioning further poses computational challenges, as perturbations span thousands of genes and complex morphological features, requiring encoding strategies that preserve interpretability while enabling cross-modal interactions Lotfollahi et al. (2023); Rampášek et al. (2022). Diffusion models add inefficiency with hundreds of denoising steps and stochastic outputs misaligned with reproducibility requirements for hypothesis-driven validation Song et al. (2020); Ni et al. (2025). Advances in spatial transcriptomics underscore the value of integrating morphology with molecular signatures at cellular resolution Moses & Pachter (2022); Rao et al. (2021), yet existing methods remain focused on tissue analysis rather than molecular design Bae et al. (2021); Williams et al. (2022).

We present Pert2Mol, the first framework for multi-modal phenotype-to-structure generation. By integrating transcriptomic and morphological features with paired control–treatment data and rectified flow dynamics, Pert2Mol enables efficient, structurally diverse molecule generation guided by complex cellular phenotypes. This establishes a paradigm for linking high-content phenotypic screening with computational hypothesis generation in drug discovery. Pert2Mol operates in the latent space of molecular autoencoders, conditioned on integrated transcriptomic and morphological embeddings. A paired transcriptome encoder with cross-attention learns gene-to-gene mappings between control and treatment states, capturing perturbation dynamics beyond differential expression. Morphological features are extracted via a ResNet encoder, providing complementary information. These embeddings condition a rectified flow transformer that learns velocity fields along straight-line trajectories from noise to molecular structures, eliminating stochastic sampling and high cost in diffusion models while enabling deterministic hypothesis generation suitable for validation.

Our contributions include: (i) The first application of rectified flow for inverse drug design using perturbation conditioning; (ii) A transformer architecture that directly models control–treatment perturbation dynamics by fusing transcriptomic and morphological signals; and (iii) Student-Teacher Self-Representation Learning (SERE), where an EMA teacher supervises student representations across depths, stabilizing training in high-dimensional multi-modal spaces. Together, these innovations enable systematic and reproducible molecule generation from phenotypic measurements, advancing phenotype-driven drug discovery.

## 2 RELATED WORK

**Generative Models for Molecular Design** have advanced from SMILES-based RNNs and VAEs Segler et al. (2018); Gómez-Bombarelli et al. (2018); Arús-Pous et al. (2019) to graph- and flow-based models such as GraphNVP, MoFlow, and GraphAF Madhawa et al. (2019); Zang & Wang (2020); Shi et al. (2020), which ensure structural validity and latent invertibility. More recently, transformers and diffusion models have improved controllability; e.g., Graph Diffusion Transformers Liu et al. (2024) integrate property encoders with a transformer denoiser for multi-conditional generation across polymers and small molecules. While these approaches optimize chemical/ physicochemical properties (e.g., solubility, synthetic accessibility), they rarely incorporate biological conditioning (transcriptomic, morphological) or perturbation dynamics. Multi-property inverse design has been explored using hierarchical VAEs, Gaussian mixture latent spaces, and encoder-decoder architectures Jin et al. (2018); Shino & Kaneko (2025); Lee & Min (2022), addressing constraints like synthetic score and gas permeability, but typically relying on static property vectors rather than high-dimensional biological readouts or paired control/ treatment states.

**Perturbation-Conditioned Molecular Design** incorporates biological data more directly. MolGene-E Ohlan et al. (2025) harmonizes bulk and single-cell transcriptomics with contrastive learning to generate molecules from perturbation-induced profiles. GexMolGen Cheng et al. (2024) produces hit-like molecules guided by gene expression signatures, and GxVAEs Li & Yamanishi (2024) predict responses from transcriptomic profiles. Yet, these methods are largely unimodal and focus on processed differential or post-treatment signatures rather than paired control-treatment modeling or joint transcriptome-morphology conditioning.

**Flow Matching and Biological Conditioning** provide deterministic generative paths and efficient sampling compared to diffusion. GraphNVP Madhawa et al. (2019) and MoFlow Zang & Wang (2020) yield invertible latent representations, but flow-based frameworks conditioned on multi-modal biological data (gene + morphology) under paired perturbation control remain unexplored. Our work addresses this gap by integrating latent rectified flow with multi-modal condition encoders to enable efficient, deterministic molecular generation from biological perturbation data.

## 3 METHODS

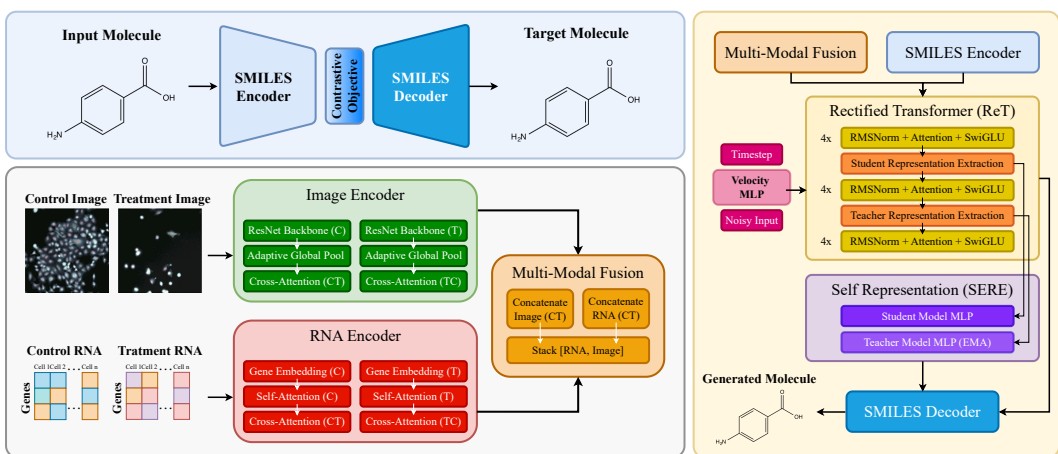

Figure 2: Overview of the Pert2Mol architecture.

We formulate drug perturbation prediction as a conditional generative modeling problem where the model learns to generate molecular structures given multi-modal biological context. Let $\mathbf{x} \in \mathbb{R}^D$ be a molecular structure in latent space, let $\mathbf{c}_{\text{img}} \in \mathbb{R}^{4 \times H \times W}$ be control and treatment microscopy images, and $\mathbf{c}_{\text{rna}} \in \mathbb{R}^G$ be gene expression profiles with $G$ genes. The objective is to learn conditional distribution $p(\mathbf{x} \mid \mathbf{c}_{\text{img}}, \mathbf{c}_{\text{rna}})$ that captures a relationship between biological perturbation context and molecular structure. Figure 2 presents the high-level architecture of Pert2Mol.

### 3.1 MULTI-MODAL CONDITIONING

**Image encoding.** To process the 4-channel microscopy images representing control and treatment conditions, we implement separate ResNet backbones for each condition. Each image encoder $f_{\text{img}}^{(C)}, f_{\text{img}}^{(T)} : \mathbb{R}^{4 \times H \times W} \to \mathbb{R}^{d_{\text{img}}}$ begins with a $7 \times 7$ convolution followed by GroupNorm, ReLU and processes through four residual layers with progressively increasing channel dimensions ($64 \to 128 \to 256 \to 512$) and spatial downsampling. Each residual block implements the skip-connection update $\mathbf{h}_{l+1} = \mathbf{h}_l + \mathcal{F}(\mathbf{h}_l, \mathbf{W}_l)$, where $\mathcal{F}$ denotes the residual function composed of two $3 \times 3$ convolutions with GroupNorm and ReLU activations. After adaptive global pooling, the condition-specific representations undergo bidirectional cross-attention:

$$\mathbf{H}'_{\text{control}} = \mathbf{H}_{\text{control}} + \text{CrossAttn}(\mathbf{H}_{\text{control}}, \mathbf{H}_{\text{treatment}}),$$

$$\mathbf{H}'_{\text{treatment}} = \mathbf{H}_{\text{treatment}} + \text{CrossAttn}(\mathbf{H}_{\text{treatment}}, \mathbf{H}_{\text{control}}).$$

The final image embeddings are produced $\mathbf{e}_{\text{img,control}}, \mathbf{e}_{\text{img,treatment}} \in \mathbb{R}^{256}$. The separate backbones enable condition-specific feature learning, while bidirectional cross-attention captures inter-condition relationships critical for understanding treatment effects on cellular morphology.

**RNA expression encoding.** For gene expression data we implement two complementary approaches for different experimental setups. In the single-condition setting, we use a standard self-attention encoder that permits genes to attend to each other based on expression patterns. The gene embedding matrix is $\mathbf{E}_{\text{gene}} = \text{Embedding}(\mathbf{c}_{\text{rna}}) \in \mathbb{R}^{G \times d_{\text{embed}}}$. For paired control–treatment experiments we implement a cross-attention mechanism between the two conditions. We first compute self-attended condition representations as:

$$\mathbf{H}_{\text{control}},\ \mathbf{H}_{\text{treatment}} = \text{SelfAttn}(\mathbf{E}_{\text{control}}),\ \text{SelfAttn}(\mathbf{E}_{\text{treatment}}),$$

and then refine each condition representation with cross-attention:

$$\mathbf{H}'_{\text{control}} = \mathbf{H}_{\text{control}} + \text{CrossAttn}(\mathbf{H}_{\text{control}}, \mathbf{H}_{\text{treatment}}),$$

$$\mathbf{H}'_{\text{treatment}} = \mathbf{H}_{\text{treatment}} + \text{CrossAttn}(\mathbf{H}_{\text{treatment}}, \mathbf{H}_{\text{control}}).$$

This cross-attention explicitly models gene-to-gene relationships between control and treatment conditions and thus captures differential expression patterns that are important for understanding drug mechanisms. Both RNA encoders use attention-weighted pooling to obtain a condition-level representation:

$$\mathbf{e}_{\text{rna}} = \sum_{i=1}^{G} \alpha_i \mathbf{h}_i, \qquad \text{where} \quad \alpha_i = \frac{\exp\big(\mathbf{w}^\top \tanh(\mathbf{W}_{\text{attn}} \mathbf{h}_i)\big)}{\sum_{j=1}^{G} \exp\big(\mathbf{w}^\top \tanh(\mathbf{W}_{\text{attn}} \mathbf{h}_j)\big)}.$$

**Multi-Modal Fusion.** Image and RNA features for each condition are concatenated and stacked to produce the final conditioning tensor:

$$\mathbf{y} = \text{Stack}\big([\mathbf{e}_{\text{img,control}} \oplus \mathbf{e}_{\text{rna,control}},\ \mathbf{e}_{\text{img,treatment}} \oplus \mathbf{e}_{\text{rna,treatment}}]\big) \in \mathbb{R}^{2 \times 192},$$

where $\oplus$ denotes concatenation. This structured representation preserves relationships between conditions while providing rich multi-modal context to the generator.

**Molecular Representation.** Molecular structures are represented using a pre-trained BERT-based autoencoder Liu et al. (2019) that maps SMILES strings to continuous latent representations similar to Chang & Ye (2024). Tokenization uses learned molecular motifs (regex tokenization) instead of character-level tokens to capture chemically meaningful substructures. A frozen pre-trained BERT encoder processes tokenized sequences, and a trainable linear compression layer reduces BERT outputs from 768 to 64 dimensions, producing a compact molecular latent representation. Concretely,

$$\mathbf{x}_{\text{target}} = \text{Linear}\big(\text{BERT}(\text{Tokenize}(\text{SMILES}))\big) \in \mathbb{R}^{64 \times 127}.$$

Continuous latent representations are amenable to gradient-based optimization; pre-trained molecular BERT captures chemical semantics, and dimensionality reduction focuses the model on task-relevant molecular features.

## 3.2 GENERATIVE MODELING

We adopt rectified flow Liu et al. (2022) as the generative framework with its training stability and sampling efficiency relative to traditional diffusion models. Rectified flow parameterizes straight-line paths between noise and data distributions. Given a noise sample $\mathbf{x}_0 \sim \mathcal{N}(0, \mathbf{I})$ and data $\mathbf{x}_1 = \mathbf{x}_{\text{target}}$, the interpolating path is $\mathbf{x}_t = (1 - t)\mathbf{x}_0 + t\mathbf{x}_1, t \in [0, 1]$, and its instantaneous velocity is given by:

$$\mathbf{v}_t = \frac{d\mathbf{x}_t}{dt} = \mathbf{x}_1 - \mathbf{x}_0.$$

The model is trained to predict this velocity field $\mathbf{v}_\theta(\mathbf{x}_t, t, \mathbf{y}) \approx \mathbf{v}_t$, with the rectified flow objective:

$$\mathcal{L}_{\text{flow}} = \mathbb{E}_{t \sim \mathcal{U}(0,1),\ \mathbf{x}_0 \sim \mathcal{N}(0, \mathbf{I})} \Big[\big\|\mathbf{v}_\theta(\mathbf{x}_t, t, \mathbf{y}) - (\mathbf{x}_1 - \mathbf{x}_0)\big\|_2^2\Big].$$

This formulation obviates complex noise scheduling, provides stable gradients compared to score-based methods, enables efficient sampling in fewer steps, and supports exact likelihood computation.

**Rectified Transformer Architecture.** Our core generative model is a transformer adapted to molecular generation with multi-modal conditioning. Noisy molecular representations are linearly projected and augmented with learnable positional embeddings: $\mathbf{X} = \text{Linear}(\mathbf{x}_t) + \mathbf{P} \in$

$\mathbb{R}^{L \times d_{\mathrm{model}}}$, and timesteps are embedded via a sinusoidal encoding passed through an MLP: $\mathbf{t}_{\mathrm{emb}} = \mathrm{MLP}(\mathrm{SinCos}(t)) \in \mathbb{R}^{d_{\mathrm{model}}}$. Conditioning information is processed along two pathways: a cross-attention pathway and an adaptive-normalization pathway:

$$\mathbf{y}_{\mathrm{cross}} = \mathrm{Linear}_{192 \to 256}(\mathbf{y}), \qquad \mathbf{y}_{\mathrm{pool}} = \mathrm{Linear}_{192 \to d_{\mathrm{model}}}\big(\mathrm{Mean}(\mathbf{y}, \dim = 1)\big)$$

The conditioning signal used for adaptive normalization is $\mathbf{c} = \mathbf{t}_{\mathrm{emb}} + \mathbf{y}_{\mathrm{pool}}$. Each transformer block implements Adaptive Layer Normalization (AdaLN) Xu et al. (2019), which modulates normalization parameters as a function of conditioning:

$$\mathrm{AdaLN}(\mathbf{x}, \mathbf{c}) = \mathbf{x} \odot \big(1 + \mathrm{scale}(\mathbf{c})\big) + \mathrm{shift}(\mathbf{c}),$$

where $\mathrm{scale}(\mathbf{c})$, $\mathrm{shift}(\mathbf{c}) = \mathrm{Linear}(\mathrm{SiLU}(\mathbf{c}))$, allowing fine-grained conditional modulation.

Each block first applies self-attention with a gating mechanism:

$$\mathbf{X}' = \mathbf{X} + \mathrm{gate}_{\mathrm{sa}}(\mathbf{c}) \odot \mathrm{MultiHeadAttn}\big(\mathrm{AdaLN}(\mathbf{X}, \mathbf{c})\big),$$

then applies cross-attention to the conditioning embedding:

$$\mathbf{X}'' = \mathbf{X}' + \mathrm{gate}_{\mathrm{ca}}(\mathbf{c}) \odot \mathrm{CrossAttn}\big(\mathrm{AdaLN}(\mathbf{X}', \mathbf{c}), \mathbf{y}_{\mathrm{cross}}\big),$$

and finally performs a feed-forward update using the SwiGLU nonlinearity:

$$\mathbf{X}_{\mathrm{out}} = \mathbf{X}'' + \mathrm{gate}_{\mathrm{ffn}}(\mathbf{c}) \odot \mathrm{SwiGLU}\big(\mathrm{AdaLN}(\mathbf{X}'', \mathbf{c})\big).$$

The SwiGLU Shazeer (2020) activation is defined as $\mathrm{SwiGLU}(\mathbf{x}) = \mathrm{SiLU}(\mathbf{x}\mathbf{W}_1) \odot (\mathbf{x}\mathbf{W}_2)\mathbf{W}_3$, and empirically provides superior performance compared to a standard MLP. AdaLN is critical for conditional generation because it provides layer-wise, conditioning-dependent modulation of activations and attention.

**Student-Teacher Self-representation (SERE).** To improve training stability and sample quality we introduce SERE. The SERE teacher model is maintained as an exponential moving average (EMA) of the student parameters: $\theta_{\mathrm{teacher}} \leftarrow \beta\,\theta_{\mathrm{teacher}} + (1 - \beta)\,\theta_{\mathrm{student}}$. During training we extract intermediate representations from selected layers: the student representation is taken at a higher-noise layer, e.g., $\mathbf{h}_{\mathrm{student}} = \mathbf{X}_{\mathrm{layer\text{-}4}}$, while the teacher representation is taken at a lower-noise layer (shifted by a small $\Delta t$), e.g., $\mathbf{h}_{\mathrm{teacher}} = \mathbf{X}_{\mathrm{layer\text{-}8}}^{\mathrm{teacher}}$. The SERE loss aligns representations using a projection head:

$$\mathcal{L}_{\mathrm{SERE}} = \big\|\mathrm{ProjectionHead}(\mathbf{h}_{\mathrm{student}}) - \mathbf{h}_{\mathrm{teacher}}\big\|_2^2,$$

with the projection head defined as

$$\mathrm{ProjectionHead}(\mathbf{h}) = \mathrm{LayerNorm}\big(\mathrm{Linear}(\mathrm{SiLU}(\mathrm{Linear}(\mathbf{h})))\big).$$

The total training objective combines the rectified flow loss and the SERE loss,

$$\mathcal{L}_{\mathrm{total}} = \mathcal{L}_{\mathrm{flow}} + \lambda_{\mathrm{SERE}}\mathcal{L}_{\mathrm{SERE}},$$

where $\lambda_{\mathrm{SERE}} = 0.1$ balances the contribution of representation alignment. SERE provides intermediate supervision beyond just the final velocity prediction, creating multiple gradient pathways through the network. Performance gain comes from the model learning what good representations should look like at different noise levels, rather than just learning to predict velocities. This representation-level supervision creates more stable, generalizable internal features that improve both training stability and sample quality shown in Figure 4.

**Training Setup.** Multi-modal data batches are assembled with temporal alignment across conditions. For classifier-free guidance, modalities are randomly masked during training: 10% RNA-only and 10% image-only inputs. SMILES strings are augmented via canonical randomization to preserve molecular identity. Optimization uses AdamW (lr=$10^{-4}$ with scheduled weight decay), mixed-precision (16-bit) for memory efficiency, gradient clipping (norm=5.0), distributed data-parallel training with synchronized gradients, and early stopping (patience=10). Two exponential moving averages are maintained: $\beta = 0.9999$ for final parameter averaging and $\beta_{\mathrm{SERE}} = 0.9999$ for updating the SERE teacher network.

Sample generation integrates the learned velocity field from noise to data. Updates follow $\mathbf{x}_{i+1} = \mathbf{x}_i + \Delta t \cdot \mathbf{v}_\theta(\mathbf{x}_i, t_i, \mathbf{y})$, using Euler integration for speed or Dormand–Prince (DOPRI5) for accuracy, with adaptive step size $\Delta t_{\mathrm{adapt}}$. Modality-specific guidance enables independent control:

$$\mathbf{v}_{\mathrm{guided}} = \mathbf{v}_{\mathrm{img}} + \lambda_{\mathrm{img}}\big(\mathbf{v}_{\mathrm{full}} - \mathbf{v}_{\mathrm{img}}\big) + \lambda_{\mathrm{rna}}\big(\mathbf{v}_{\mathrm{rna}} - \mathbf{v}_{\mathrm{img}}\big),$$

where $\mathbf{v}_{\text{full}}$, $\mathbf{v}_{\text{img}}$, and $\mathbf{v}_{\text{rna}}$ are predictions conditioned on both, image-only, and RNA-only inputs, respectively. Latent molecular representations are decoded to SMILES with our pre-trained autoencoder and beam search to improve recovery and validity. The generative model is a 12-layer transformer (768 hidden dims, 16 attention heads) implemented in PyTorch, trained for 200 epochs with batch size 64 on 4 NVIDIA H100 GPUs. Experiments use GDP, LINCS RNA, CPGJump datasets. Evaluation metrics include molecular validity, uniqueness, and biological relevance.

## 4 EXPERIMENT

**Datasets.** We build upon the multi-modal Ginkgo Data Platform (GDP) collection Model & Biologics (2025), which pairs transcriptomic profiles with four-channel fluorescence microscopy of chemically perturbed cell populations. Correspondence across modalities was established through compound identity matching and experimental parameter alignment, with metadata standardized for dose normalization, consistent cell line identifiers, and synchronized timepoints; DMSO-treated samples served as controls. Transcriptomic data were normalized to $10^6$ reads per sample, log1p transformed, and reduced to the top 2000 variable genes via scanpy. Imaging data were scaled from 16-bit to unit range, contrast-adjusted per channel using percentile clipping, and resampled with bilinear interpolation. This framework preserved the full combinatorial design (cell line, drug-dose, timepoint) during partitioning, enabling controlled comparisons between compounds and vehicle controls. Canonical SMILES for PubChem drugs Wang et al. (2009) were obtained, with up to 20 variants enumerated to train molecular autoencoders; evaluation outputs were standardized into canonical SMILES with RDKit Landrum (2016). For single-modality experiments, we trained on GDP RNA-seq and Cell Painting data, and further incorporated preprocessed LINCS gene expression data Subramanian et al. (2017) covering >3000 PubChem-matched perturbations. To mitigate LINCS batch effects Hetzel et al. (2022), we applied Harmony Korsunsky et al. (2019) for dimension-matched normalization. Since our model uses both pre- and post-treatment data, it learns treatment-induced changes even under Harmony transformation, whereas simple vector differences after correction would be invalid. This design allows leveraging batch-effect corrections under uniform transformations. We also evaluated on Cell Painting (cpgjump-pilot) Chandrasekaran et al. (2024), comprising ∼300 compounds across two cell lines, for imaging-only experiments.

**Evaluation Metrics.** We evaluate molecular generation for drug discovery Grant & Sit (2021) using a diverse set of metrics. Validity ensures chemically sound structures, while the Fréchet ChemNet Distance (FCD) quantifies similarity to the target dataset by comparing deep neural network activations that capture chemical and biological properties. Quantitative Estimation of Drug-likeness (QED) provides a unified drug-likeness score, and Lipinski compliance measures adherence to the Rule of Five, critical for oral bioavailability. We further assess the drug-like fraction and target similarity to capture therapeutic relevance. Finally, KL divergences for molecular weight, LogP, and TPSA quantify preservation of key physicochemical property distributions. Together, these metrics provide a comprehensive evaluation of validity, drug-likeness, therapeutic relevance, and property distribution preservation, satisfying core requirements for practical drug discovery.

**Generation Task.** Pert2Mol demonstrates the capacity for de novo molecular design by generating novel drug candidates directly conditioned on biological perturbation profiles. The model employs a rectified flow-based generative process to sample molecular structures from learned distributions, conditioned on encoded pre- and post-treatment biological features. This capability directly measures Pert2Mol's fundamental generative abilities and its capacity to explore chemically meaningful regions of molecular space while maintaining therapeutically relevant properties.

**Repurposing Task.** Pert2Mol can perform drug selection guided by known-context for unseen (test) data through biological similarity matching. Here we encode test biological conditions using the same feature extraction pipeline as the generative framework, then performs nearest neighbor search in the known biological feature space from provided training data, to identify similar conditions from the training set using cosine similarity metrics. The drugs associated with these nearest biological profiles are returned as predictions. Performance is quantified using same metrics as generation evaluation. This capability establishes how Pert2Mol's learned representations compare against direct biological similarity-based approaches for therapeutic compound identification.

**Retrieval Task.** To show meaningful learned drug representations, Pert2Mol performs retrieval accuracy analysis on augmented train data. Each training sample is used as a query to retrieve nearest

neighbors in the 192-dimensional embedding space using cosine similarity, testing whether samples treated with the same compound clustered together. Retrieval performance was measured with Precision@K and Hit@K (K=1,3,5,10), Mean Reciprocal Rank (MRR), and average intra-compound similarity. Clustering quality was further assessed via separation scores (intra- vs. inter-compound similarity), identifying compounds most effectively distinguished by the learned embeddings. This framework validates whether the RNA-image encoder captures compound-specific biological effects, a prerequisite for downstream tasks such as mechanism-of-action prediction and repurposing.

## 4.1 RESULTS

Since no existing method tackles the task of perturbation guided drug molecule design, we compared our method with a diffusion-based baseline. We also trained our model with only RNA-seq or CellPainting imaging input, as well as with a version with SERE disabled as ablations.

Pert2Mol achieves superior molecular generation performance across all evaluation metrics. On the GDP dataset (Table 1), the full multi-modal model attains an FCD score of 4.996, outperforming the diffusion baseline (7.343) and all ablation variants. The model maintains perfect molecular validity (1.0) while achieving a QED score of 0.587 and Lipinski compliance rate of 78.5%, compared to the baseline's 55.2% QED and 71.5% Lipinski compliance. The model demonstrates precise physicochemical property distribution preservation with KL divergences of 0.263 for molecular weight, 0.195 for LogP, and 0.208 for TPSA, substantially lower than baseline values of 1.524, 1.044, and 3.081 respectively. Among single-modality variants, Pert2Mol-RNA (FCD: 5.501) outperforms Pert2Mol-image (FCD: 5.762), indicating transcriptomic features provide more informative signals than imaging features alone. Cross-dataset evaluation (Table 2) shows consistent performance with FCD scores of 4.996 (GDP), 2.748 (CPG-jump), and 8.354 (LINCS). All datasets maintain perfect validity while achieving QED scores between 0.551-0.587 and Lipinski compliance rates of 76.7-79.6%.

Table 1: Molecular generation performance on GDP dataset, compared against baseline & ablations.

| Method | Validity | FCD↓ | QED↑ | Lipinski↑ | Drug-like↑ | Target Sim.↑ | MW KL↓ | LogP KL↓ | TPSA KL↓ |
|---|---|---|---|---|---|---|---|---|---|
| Baseline | 0.962 | 7.343 | 0.552 | 0.715 | 0.557 | 0.133 | 1.524 | 1.044 | 3.081 |
| Pert2Mol-RNA | 1.0 | 5.501 | 0.568 | 0.723 | 0.591 | 0.136 | 0.827 | 0.870 | 0.819 |
| Pert2Mol-image | 0.998 | 5.762 | 0.538 | 0.702 | 0.522 | 0.136 | 0.332 | 0.249 | 0.41 |
| Pert2Mol-wo-SERE | 1.0 | 6.809 | 0.582 | 0.745 | 0.627 | **0.143** | 0.153 | 0.274 | 0.282 |
| Pert2Mol | 1.0 | **4.996** | **0.587** | **0.785** | **0.652** | 0.136 | **0.263** | **0.195** | **0.208** |

Table 2: Molecular generation performance on all datasets for Pert2Mol model.

| Dataset | Validity | FCD↓ | QED↑ | Lipinski↑ | Drug-like↑ | Target Sim.↑ | MW KL↓ | LogP KL↓ | TPSA KL↓ |
|---|---|---|---|---|---|---|---|---|---|
| GDP | 1.0 | 4.996 | 0.587 | 0.785 | 0.652 | 0.136 | 0.263 | 0.195 | 0.208 |
| CPG-jump | 1.0 | 2.748 | 0.551 | 0.767 | 0.611 | 0.103 | 0.159 | 0.091 | 0.167 |
| LINCS | 1.0 | 8.354 | 0.568 | 0.796 | 0.621 | 0.104 | 0.052 | 0.062 | 0.096 |

Drug repurposing evaluation results in Table 3 reveals Tanimoto similarity scores of $0.571 \pm 0.427$ for the full model on GDP dataset, compared to $0.498 \pm 0.411$ for the baseline. Pert2Mol-RNA achieves the highest similarity score of $0.597 \pm 0.415$, while Pert2Mol-image ($0.407 \pm 0.378$) and Pert2Mol-wo-SERE ($0.401 \pm 0.371$) show reduced performance, highlighting the importance of transcriptomic data and cross-modal integration mechanisms. Cross-dataset analysis in Table 4 shows dataset-dependent performance with GDP achieving $0.571 \pm 0.427$, while CPG-jump and LINCS datasets yield lower similarity scores of $0.195 \pm 0.164$ and $0.198 \pm 0.154$ respectively, despite maintaining high Lipinski compliance rates (95.5% and 99.8%).

Compound retrieval analysis on GDP dataset (Table 5) demonstrates the effectiveness of learned drug representations. Pert2Mol achieves Precision@1 of 0.7610 and Hit@1 of 0.7610, compared to 0.6560 for both metrics in the SERE-disabled variant. Performance scales consistently across retrieval depths, reaching Precision@10 of 0.7659 and Hit@10 of 0.9690. The Mean Reciprocal Rank increases from 0.7656 (without SERE) to 0.8364 (full model), confirming the critical role of cross-condition attention mechanisms. The validation loss curves (Figure 4) reveal distinct convergence patterns between model variants. Pert2Mol achieves rapid convergence with validation loss decreasing from 0.72 to approximately 0.10 by step 2500, while Pert2Mol-no-SERE shows slower

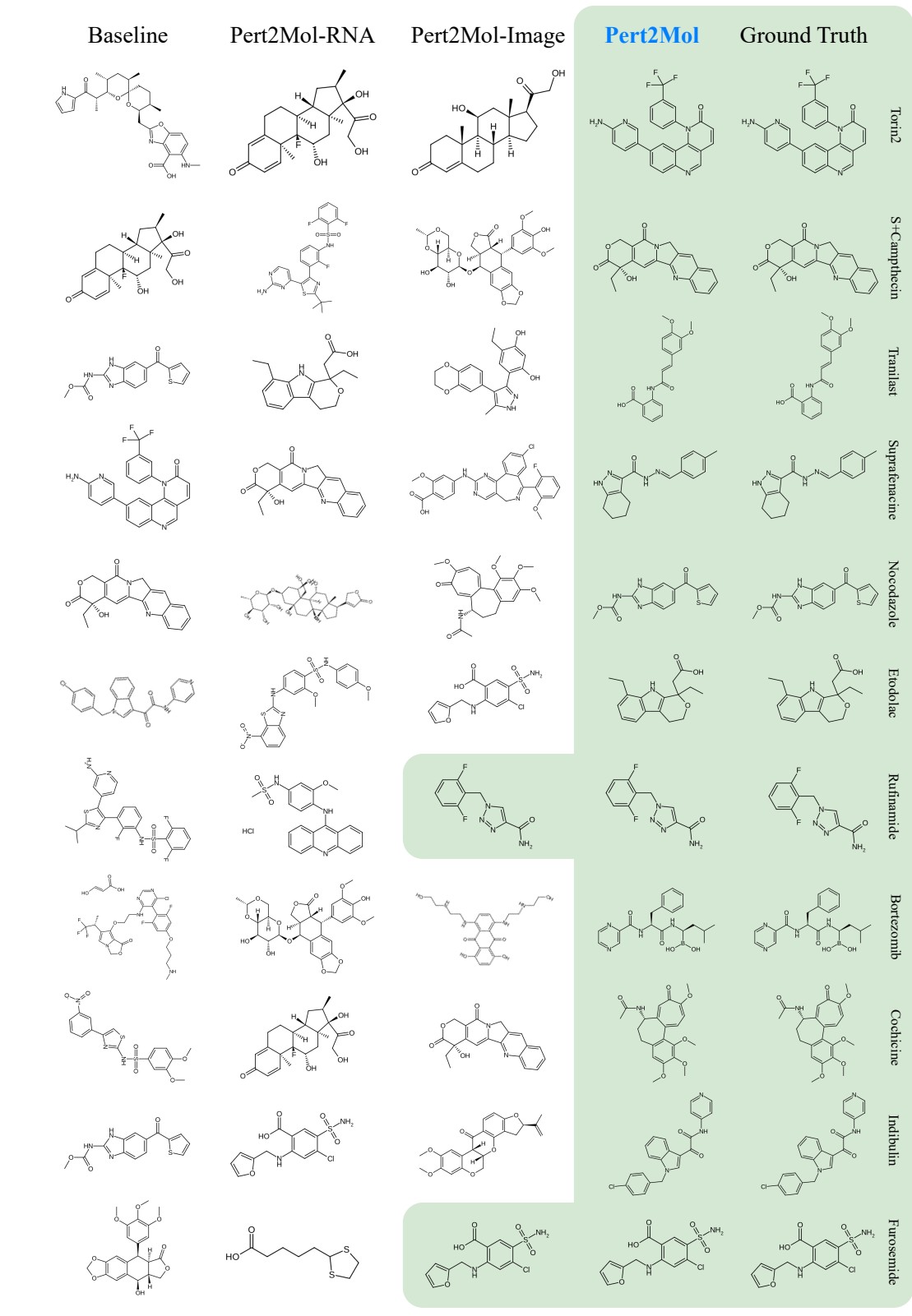

Figure 3: Molecule generation results for given control-treatment RNA-image input for different drug examples from GDP dataset. Our Pert2Mol is compared against diffusion baseline along with RNA & image only models.

Table 3: Repurposing performance on GDP dataset, compared against baseline & ablations.

| Method | Validity | QED↑ | Lipinski↑ | Tanimoto Sim.↑ |
|---|---|---|---|---|
| Baseline | 1.0 | $0.537 \pm 0.206$ | 0.824 | $0.498 \pm 0.411$ |
| Pert2Mol-RNA | 1.0 | $0.536 \pm 0.193$ | 0.750 | $\mathbf{0.597 \pm 0.415}$ |
| Pert2Mol-image | 1.0 | $0.407 \pm 0.378$ | 0.878 | $0.407 \pm 0.378$ |
| Pert2Mol-wo-SERE | 1.0 | $0.550 \pm 0.205$ | 0.785 | $0.401 \pm 0.371$ |
| Pert2Mol | 1.0 | $\mathbf{0.552 \pm 0.213}$ | **0.878** | $0.571 \pm 0.427$ |

Table 4: Repurposing performance on all datasets

| Dataset | Validity | QED↑ | Lipinski↑ | Tanimoto Sim.↑ |
|---|---|---|---|---|
| GDP | 1.0 | $0.552 \pm 0.213$ | 0.878 | $0.571 \pm 0.427$ |
| CPG-jump | 1.0 | $0.569 \pm 0.192$ | 0.955 | $0.195 \pm 0.164$ |
| LINCS | 1.0 | $0.561 \pm 0.168$ | 0.998 | $0.198 \pm 0.154$ |

Figure 4: SERE ablation

convergence, declining from 0.95 to 0.43 over the same period. The full model demonstrates superior optimization dynamics with lower final loss values, supporting the quantitative performance improvements observed across all evaluation tasks.

Table 5: Retrieval performance on GDP dataset

| Metric | Pert2Mol-wo-SERE | | Pert2Mol | |
|---|---|---|---|---|
| | Precision | Hit | Precision | Hit |
| @1 | 0.6560 | 0.6560 | 0.7610 | 0.7610 |
| @3 | 0.6730 | 0.8590 | 0.7653 | 0.8920 |
| @5 | 0.6766 | 0.9030 | 0.7640 | 0.9350 |
| @10 | 0.6840 | 0.9330 | 0.7659 | 0.9690 |
| MRR | 0.7656 | | 0.8364 | |

## 5 DISCUSSION

Pert2Mol establishes the first successful framework for multi-modal phenotype-to-structure generation. We demonstrate that integrating transcriptomic and morphological features can effectively bridge the gap between complex biological perturbation data and molecular design. Our latent rectified flow method consistently outperforms diffusion-based baseline approaches across drug discovery metrics while maintaining perfect molecular validity. Through SERE our model learns enhanced representations without additional components, showing training stability, which is crucial for generative modeling in high-dimensional multi-modal spaces. The bidirectional cross-attention mechanism successfully captures perturbation dynamics beyond simple differential measurements, enabling direct learning from paired control-treatment experimental data rather than preprocessed differential vectors.

While our method currently requires paired control-treatment data, it could be adapted for targeted perturbations of specific pathways or phenotypic features. Performance variability across datasets reflects differences in measurement technologies, presenting opportunities to explore unified datasets such as large-scale single-cell RNA-seq screening where foundational models could standardize data curation. Although our framework generates chemically valid structures with favorable drug-like properties, experimental validation of biological activity remains essential for translating computational predictions into therapeutic applications. Despite these limitations, Pert2Mol represents a significant step forward in systematic, reproducible molecule generation from phenotypic measurements, offering a new approach for phenotype-driven drug discovery that could accelerate the identification of novel therapeutic compounds from high-content screening data.

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

# A APPENDIX

## A.1 ETHICS STATEMENT

Large language models served only as proofreading and editorial assistance tools for the manuscript text, with no participation in data analysis, scientific interpretation, or content development.

## A.2 REPRODUCIBILITY STATEMENT

Code and pretrained model weights will be released upon acceptance.

