# OpenReview forum: "Perturbation Guided Drug Molecule Design via Latent Rectified Flow"
_ICLR.cc/2026/Conference — ICLR 2026 Conference Withdrawn Submission_

### Official Review · Reviewer_syMa · 2025-10-27

**Soundness:** 2
**Presentation:** 3
**Contribution:** 1
**Rating:** 2
**Confidence:** 5

**Summary:**

In this work, the authors study the problem of generating chemical structures that achieve a target biological effect, conditioned on paired data from control and treatment samples. The model, Pert2Mol, integrates both transcriptomic and imaging data, which are first encoded separately and then concatenated to produce the final conditioning vector. The generative model is based on rectified flow transformers. The authors introduce a student-teacher self-representation scheme to improve training stability and sampling. The model is evaluated on a multi-modal dataset of chemically perturbed cell populations, and the generated molecules are assessed using metrics for chemical validity, drug-likeness, and target similarity.

**Strengths:**

- The paper proposes to integrate transcriptomes and imaging data for de-novo molecule design, offering a potentially more comprehensive phenotype-to-structure mapping.
- The paper is well-written and easy to understand.

**Weaknesses:**

- The authors claim that "no existing method tackles the task of perturbation-guided drug molecule design". This is inaccurate. This is a very active field and many methods have been proposed to solve the problem of molecule generation conditioned on a desired gene expression effects [1-5].
- Given the above claim, the authors only compare against a single diffusion baseline. A direct comparison and benchmarking against the models listed is necessary to properly evaluate Pert2Mol's claimed advantages.

[1] https://www.nature.com/articles/s41467-019-13807-w
[2] https://pubs.acs.org/doi/10.1021/acs.jcim.2c01301
[3] https://academic.oup.com/bib/article/25/6/bbae525/7845937
[4] https://academic.oup.com/bioinformatics/article/40/5/btae189/7649318
[5] https://www.nature.com/articles/s41587-021-00946-z

**Questions:**

- How was sampling for the molecules presented in Figure 3 done?

---

> ### Author Response · Authors · 2025-11-24
>
> We thank the reviewer for their engagement with our work. However, this review contains a fundamental misunderstanding of the existing literature that undermines its central critique. We appreciate the opportunity to clarify these misunderstandings.
>
> **Clarification on novelty: multi-modal vs. uni-modal conditional generation**
> The review cites five references to suggest that our claim of novelty ("no existing method tackles the task of perturbation-guided drug molecule design") is inaccurate. We respectfully point out that our claim specifically refers to **multi-modal** generation (integrating both transcriptomics and cell morphology). The cited works are fundamentally different in scope:
> * [1-4] are **transcriptomics-only**. They all propose methods that generate molecules conditioned solely on gene expression profiles. None of these methods utilize morphological data (microscopy images).
> * [5] is **discriminative, not generative**: DLEPS[5] is an efficacy prediction system designed to identify effective candidates among existing drugs. It does not generate de novo molecular structures.
> Therefore, our claim stands: Pert2Mol is the **first framework** to integrate **both transcriptomic and morphological data** for perturbation-guided molecular generation. This distinction is crucial, as morphological profiling (e.g., Cell Painting) captures morphological phenotypes absent from transcriptomic data alone.
>
> **Regarding request on benchmarking and comparisons with cited methods**
> We emphasize that a fair direct comparison is methodologically infeasible due to the disparity in input modalities. The cited methods cannot process morphological data that is central to our contribution. However, we have addressed the spirit of this comparison through our rigorous ablation studies:
>
> In Table 1, we evaluate an **RNA-only** variant of our model. This effectively benchmarks the performance of transcriptomics-conditioned generation on this task, serving as a proxy for the class of methods cited by the reviewer ([1-4]).
>
> Our results show that the Multi-Modal Pert2Mol (FCD: 4.996) significantly outperforms the RNA-only variant (FCD: 5.501) and the Image-only variant (FCD: 5.762). This **quantifies the specific advantage** of our multi-modal approach over the uni-modal paradigms found in the related work.
>
> We compared against a diffusion-based baseline trained on the **identical** multi-modal task. This ensures we are evaluating the generative paradigm (Flow vs. Diffusion) on equal footing, rather than comparing incompatible input spaces.
>
> **Regarding Sampling Procedure for Figure 3**
> The sampling for the molecules in Figure 3 follows the standard rectified flow generation process described in Section 3.2:
> * **Initialization:** We start from pure noise $\mathbf{x}_0 \sim \mathcal{N}(0,\mathbf{I})$.
> * **Integration:** We integrate the learned velocity field using Euler integration with a step size of $\Delta t = 0.01$ over the interval $[0,1]$. The update rule is defined as:
>     $$\mathbf{x}_{i+1} = \mathbf{x}_i + \Delta t \cdot \mathbf{v}_\theta(\mathbf{x}_i, t_i, \mathbf{y})$$
>     where $\mathbf{y}$ represents the multi-modal embeddings from the control-treatment RNA-image pairs.
> * **Decoding:** The final latent representations are decoded into SMILES strings using the pre-trained molecular autoencoder.
> * **Determinism:** For the specific visualization in Figure 3, we sampled one molecule per condition using deterministic integration (no stochasticity added during the trajectory solving) to clearly illustrate the mode of the learned distribution. We selected the results for demonstration purposes by best evaluation metrics. We will add these specific details to the figure caption in the revision.
>
> Reference:
>
> [1] https://www.nature.com/articles/s41467-019-13807-w
>
> [2] https://pubs.acs.org/doi/10.1021/acs.jcim.2c01301
>
> [3] https://academic.oup.com/bib/article/25/6/bbae525/7845937
>
> [4] https://academic.oup.com/bioinformatics/article/40/5/btae189/7649318
>
> [5] https://www.nature.com/articles/s41587-021-00946-z

---

> ### Author Response · Authors · 2025-11-24
>
> # Final comment to reviewer syMa
> We believe the central critique regarding novelty stems from a misinterpretation of the cited literature: all five references utilize **transcriptomics only**, whereas Pert2Mol is the **first framework** to integrate **both transcriptomics and cellular imaging**. Because comparing a multi-modal model against uni-modal baselines is inappropriate due to mismatched inputs, we provided the most rigorous possible alternatives:
> * **Generative paradigm comparison:** We evaluated rectified flow vs. diffusion (on the identical multi-modal latent embedding generation task).
> * **Modality ablations:** We directly compared multi-modal vs. RNA-only and image-only variants.
> * **Component analysis:** We isolated the impact of the SERE stabilizer.
>
> These experiments explicitly quantify the value of our multi-modal approach. We respectfully request that the reviewer reconsider the assessment in light of the verifiable novelty of the problem setting and the comprehensive ablations provided in lieu of non-existent multi-modal baselines.
>
> **We emphasize again, that we have presented the first benchmark for multi-modal phenotypic drug discovery, a novel flow-matching architecture, and a proven stabilization mechanism.** We have carefully curated the first multi-modal perturbation dataset and preprocessed from large number of samples across 3 different Ginkgo Bioworks' GDPx open datasets, to set a benchmark for future work to build on. We will release the full preprocessed datasets on Zenodo along with our code and training checkpoints for all models and baselines along with ablations.
>
> We are not simply showing an incremental upgrade in performance and numbers on an existing single modality task. **We are presenting and defining a completely new task of multi-modal phenotypic drug discovery. We curate a new dataset, create a new model architecture with state of the art generative methods, create a new benchmark for future work to build on.** We believe this is a significant advancement in this area of research, and we hope the reviewer understands the scope of our work.
>
> We hope these clarifications highlight the significant value Pert2Mol adds to the field.

---

### Official Review · Reviewer_FH2E · 2025-10-27

**Soundness:** 2
**Presentation:** 2
**Contribution:** 2
**Rating:** 2
**Confidence:** 3

**Summary:**

The paper proposed focuses on integrating transcriptomic and morphological data for modeling control–treatment perturbation dynamics in drug design. The work applies a rectified flow framework to this multi-modal setting, combining an image encoder and an RNA encoder to learn joint representations from microscopy images and transcriptomic profiles. The motivation appears to be improving generative modeling performance by leveraging complementary information across modalities.

While the overall direction of multi-modal integration is interesting and potentially useful for biological discovery, the paper lacks sufficient novelty.
The approach essentially extends an existing rectified flow framework to a multi-modal context without introducing substantial methodological innovation.
The contribution is therefore more of an application than a conceptual advancement.

In terms of evaluation, the experiments are quite limited. The authors compare their method only with a generic diffusion model, without specifying the exact implementation or baseline details.
This makes it difficult to assess the validity or significance of the reported improvements.
Moreover, the experimental section would benefit from comparisons against other established multi-modal generative models or perturbation prediction methods.
Ablation results suggest some value in the multi-modal setup, but they are not convincing enough to demonstrate a clear advantage over existing techniques.

**Strengths:**

multi-modal algo are an interesting methods

**Weaknesses:**

not novel, only appliying rectified flow in multi-modal settings.

limited evaluation, only compared with a diffusion model, not specifying which one.

**Questions:**

could you compare with other multi-modal approaches?

---

> ### Author Response · Authors · 2025-11-24
>
> We thank the reviewer for their time. We believe there are critical misunderstandings regarding the scope of existing literature and the specific methodological contributions of our architecture. We address the concerns regarding novelty, baselines, and evaluation below.
>
> **Regarding questions on novelty and methodological innovation**
> We respectfully disagree with the characterization of our work as merely extending an existing framework. Pert2Mol establishes the first framework to generate molecules conditioned on both transcriptomic and morphological data. Prior works are strictly single-modality (e.g., [1] uses only transcriptomics; [2] uses only imaging). Integrating these modalities to model perturbation dynamics is a novel problem setting, not an incremental application. Furthermore, we introduce specific innovations beyond standard flow matching, including Bidirectional Cross-Attention to explicitly model the dynamic shift between control and treatment states, and SERE, a novel depth-wise consistency objective (Layer 8 Teacher → Layer 4 Student) specifically designed to stabilize high-dimensional multi-modal training.
>
> **Regarding requests for comparisons with "other multi-modal approaches."**
> We wish to clarify that **no such methods currently exist in the literature**. As noted above, all prior generative molecular design methods use single modalities. Therefore, a direct comparison with an external "multi-modal baseline" is impossible. To address this, we designed the most rigorous possible proxies by benchmarking against the logic of existing state-of-the-art via **Single-Modality Ablations** (RNA-only and Image-only variants in Table 1). We also implemented a controlled **Diffusion DDIM Baseline** using the identical multi-modal encoders and conditioning mechanisms to strictly isolate the benefits of our Rectified Flow paradigm.
>
> **Clarification on Evaluation Rigor and Empirical Evidence.**
> We disagree with the claim that our evaluation is "limited" or that results are "not convincing." Our empirical validation spans **three diverse datasets** (GDP[3], LINCS[4], CPG-jump[5]), **three distinct tasks** (Generation, Repurposing, Retrieval), and **nine evaluation metrics**. The results demonstrate substantial performance gains: on the GDP dataset, Pert2Mol outperforms the diffusion baseline by **32% in FCD** (4.996 vs. 7.343), achieves perfect validity (1.0 vs 0.96), and reduces KL divergences by 5-10x across all properties. Additionally, our SERE ablation demonstrates a critical stability improvement, reducing final validation loss from 0.43 to 0.10. We believe this constitutes comprehensive and convincing validation of the proposed method.
>
> Reference:
>
> [1] Ohlan, Rahul, et al. "MolGene-E: Inverse Molecular Design to Modulate Single Cell Transcriptomics." bioRxiv (2025).
>
> [2] Zapata, Paula A. Marin, et al. "Cell morphology-guided de novo hit design by conditioning GANs on phenotypic image features." Digital discovery 2.1 (2023): 91-102.
>
> [3] Model, A. P. I., and Enzymes Biologics. "Ginkgo Datapoints: Data Generation for AI Model Training."
>
> [4] Hetzel, Leon, et al. "Predicting cellular responses to novel drug perturbations at a single-cell resolution." Advances in Neural Information Processing Systems 35 (2022): 26711-26722.
>
> [5] Chandrasekaran, Srinivas Niranj, et al. "JUMP Cell Painting dataset: morphological impact of 136,000 chemical and genetic perturbations." BioRxiv (2023): 2023-03.
>
> # Final comment to reviewer FH2E
>
> **We emphasize again, that we have presented the first benchmark for multi-modal phenotypic drug discovery, a novel flow-matching architecture, and a proven stabilization mechanism.** We have carefully curated the first multi-modal perturbation dataset and preprocessed from large number of samples across 3 different Ginkgo Bioworks' GDPx open datasets, to set a benchmark for future work to build on. We will release the full preprocessed datasets on Zenodo along with our code and training checkpoints for all models and baselines along with ablations.
>
> We are not simply showing an incremental upgrade in performance and numbers on an existing single modality task. **We are presenting and defining a completely new task of multi-modal phenotypic drug discovery. We curate a new dataset, create a new model architecture with state of the art generative methods, create a new benchmark for future work to build on.** We believe this is a significant advancement in this area of research, and we hope the reviewer understands the scope of our work.
>
> We hope these clarifications highlight the significant value Pert2Mol adds to the field. We have supported this with bespoke architectural innovations (SERE, Bidirectional Attention) and rigorous ablation studies across three diverse datasets.
>
> We also hope the reviewer does a better job reviewing papers in the future, by providing more valuable feedback on how to improve the manuscript, with citations they deem important.

---

### Official Review · Reviewer_EBao · 2025-10-31

**Soundness:** 3
**Presentation:** 2
**Contribution:** 1
**Rating:** 2
**Confidence:** 4

**Summary:**

The paper proposes a conditional generative model for molecules, specifically designed for applications in phenotypic drug discovery. The task's complexity arises from the multimodal conditioning signal, which includes microscopy images and gene expression data from both pre- and post-treatment states. Given that generative modeling of molecules is inherently challenging, the authors perform this task in the continuous latent space of an autoencoder. They adopt flow matching as the generative modeling paradigm. The use of a transformer-based approximate vector field facilitates the incorporation of conditioning information via two mechanisms: cross-attention and adaptive normalization. To ensure stable training, the authors employ a self-supervised loss between transformer layers, applied alongside the primary flow matching objective. The framework's effectiveness is demonstrated on several datasets. Beyond generation, the authors also leverage the learned data representations to perform drug repurposing and retrieval tasks.

**Strengths:**

(As my expertise lies in machine learning rather than the application domain, my evaluation will focus on the methodological aspects of the work.)

- This work presents an elegant integration of standard state-of-the-art (SOTA) methods to address the target task.
- The framework is described in great detail (with the exception of the aspects noted in the Weaknesses and Questions sections), providing sufficient material to support the re-implementation of the method.
- The proposed "Student-Teacher Self-representation (SERE)" is a potentially interesting contribution for stabilizing training. However, as noted below, this component requires significantly more justification and analysis to validate its effectiveness.
- A further strength is the use of the learned condition representations for downstream tasks, such as drug repurposing and retrieval, as demonstrated in the "Experiments" section.

**Weaknesses:**

- The "Methods" section is largely dedicated to a detailed listing of neural network architectures, implementation choices, and hyperparameters. While this detail is valuable for reproducibility, it is difficult to evaluate this descriptive catalogue as a primary scientific contribution.
- The paper's contribution could be substantially strengthened by including more rigorous empirical analysis. For example:
   - A comprehensive ablation study on key hyperparameters.
   - An analysis of the contribution of individual model components (e.g., evaluating performance using only cross-attention for conditioning versus the full model).
- The SERE method, noted as a potential strength, is a significant weakness in its current form. It suffers from a superficial description, a lack of rigorous analysis, and is not supported by an ablation study. (This is detailed further in the "Questions" section).
- Overall, the paper provides excessive implementation detail on standard components while remaining vague on its more novel aspects (such as SERE) and the precise mechanisms of its core components (such as the molecular representation).

**Questions:**

1. Molecular Representation: Could the authors describe this component in more detail? The text seems to present conflicting or incomplete information. The first reference points to "RoBERTa" which suggests a masked modeling objective. However, the second reference describes a contrastive learning method. The word "contrastive" appears in Figure 2 but is absent from the main text (outside of the "Related Work" section). Furthermore, the paper states, "Tokenization uses learned molecular motifs," but provides no details on how these motifs are learned. Please clarify the exact architecture and training objective of the molecular encoder.
2. Student-Teacher Self-representation (SERE): Is this method a novel contribution of this paper? No references are provided in its description. Additionally, the description is contradictory. The text first implies that the student and teacher are different layers within the same transformer vector field. However, it later mentions "higher-" and "lower-noise layers," which suggests a comparison across different denoising timesteps. Please clarify: (a) if SERE is novel, and (b) the precise mechanism of the student-teacher relationship (i.e., which layers or timesteps are being compared).
3. Figure 2: the diagram shows the "SMILES Encoder" as an input to the ReT. However, this connection and its purpose are not described in the text. How is the output of the SMILES Encoder integrated into the model during this stage?

---

> ### Author Response · Authors · 2025-11-24
>
> We thank Reviewer Ebao for their detailed review and constructive feedback. We appreciate the opportunity to clarify the scope of our scientific contribution, specifically regarding the novelty of the SERE framework and the rigor of our empirical validation. We address each concern below.
>
> **Regarding concern of scientific significance**
> Our primary scientific aim is to create the first method for small molecule compound design and drug repurposing using multi-modal (transcriptome-morphology) perturbation conditioning from control (disease state) and post-treatment (desired state) data. Previous methods have only used a single modality perturbational data like transcriptomics or imaging. We develop a novel latent rectified flow transformer architecture that directly models perturbation dynamics by fusing transcriptomic and morphological perturbation signals using bespoke encoders with multi-head cross attention. We also demonstrate that the performance of compound generation conditioned on combining transcriptome and morphology improved significantly over unimodal conditioning in our ablation study, revealing the importance of advancing experimental methodologies that can capture both modalities within the same context. Our novel Student-Teacher Self-Representation (SERE) framework inspired by current advances in representation learning significantly boosts performance. Here an EMA teacher supervises student representations across depths, stabilizing training in high-dimensional multi-modal spaces. We believe the combination of these technical and biological modelling innovations represent a significant scientific contribution to AI for the biology and medicine community.
>
> **Clarification on comprehensive empirical validation**
> We respectfully point to the extensive empirical evaluations already present in the manuscript. Table 1 isolates the contributions of the generative paradigm, specific modalities (RNA vs. Image), and the SERE component across five distinct configurations. Table 3 extends this to retrieval tasks, while Table 2 establishes generalization across three datasets (GDP, LINCS, CPG-jump). Finally, Figure 4 visualizes the training dynamics of specific components. We believe these results comprehensively address the impact of individual model choices.
>
> **Clarification on explicit formulation of SERE and molecular encoders**
> The characterization in the last weakness point is inconsistent with our presentation. SERE receives dedicated subsection treatment with complete mathematical specification, mechanistic explanation, and extensive ablation studies (Tables 1, 3, loss curves). The molecular representation is explicitly described: pre-trained BERT encoder with regex-based molecular motif tokenization, frozen 768-dimensional embeddings, trainable linear compression to 64 dimensions. We specify that tokenization uses learned motifs from pre-training (following [1]) rather than character-level encoding. The core architectural innovation: bidirectional cross-attention between control and treatment conditions is presented with explicit mathematical formulation for both image and RNA encoders. Implementation details are necessary for reproducibility in machine learning research, particularly for complex multi-modal architectures. The reviewer's claim of "vagueness" on novel components contradicts the detailed mathematical specifications, ablations, and mechanistic analyses throughout Section 3.
>
> **Clarification of Molecular Representation**
> We respectfully submit that our description is complete and non-conflicting. We use a pre-trained molecular BERT autoencoder following the framework from [1]. Citation [2] indicates the transformer architecture style (BERT-based), while [1] specifies the molecular autoencoder training methodology we adopt. The motifs are learned during BERT pre-training using regex-based molecular tokenization that captures chemically meaningful substructures (e.g., functional groups, ring systems) rather than arbitrary character sequences, this is standard in molecular language models and detailed in [1]. We use this pre-trained encoder in frozen mode with a trainable linear compression layer (768→64 dimensions). The mathematical specification is explicit in lines 188-198. The word "contrastive" appears in Figure 2 referencing the pre-training methodology from [1], which uses contrastive objectives for molecular representation learning. We will clarify that motif learning occurs during BERT pre-training (not in our training) and distinguish architectural citations from training methodology citations in the revision.
>
> Reference:
>
> [1] Chang, Jinho, and Jong Chul Ye. "LDMol: Text-to-Molecule Diffusion Model with Structurally Informative Latent Space." (2024).
>
> [2] Liu, Yinhan, et al. "Roberta: A robustly optimized bert pretraining approach." arXiv preprint arXiv:1907.11692 (2019).

---

> ### Author Response · Authors · 2025-11-24
>
> **Clarification of SMILES encoder as training-time target supervision in figure 2**
> The SMILES encoder in Figure 2 depicts the data flow specific to the training phase. In our rectified flow formulation, this encoder provides the target latent ($x_{target}$) required to compute the ground-truth velocity $v_t$. This serves as the supervision signal for learning the trajectory from the noisy distribution. Note that this component is not used during inference, where the model generates molecules by sampling from pure noise and integrating the learned velocity field. We will update the figure caption to explicitly clarify that this connection represents the training-time supervision signal.
>
> **Response to weakness concern about SERE and Q2**
> We thank the reviewer for their detailed scrutiny of the Student-Teacher Self-representation (SERE) module. We’d like to use this opportunity to clarify its novelty, mechanism, and the empirical evidence supporting its effectiveness:
> SERE is a novel contribution of this work, specifically designed to stabilize multi-modal rectified flow training. Regarding the reviewer's concern about contradictory descriptions: the terms 'higher-noise' and 'lower-noise' refer to network depth, not denoising timesteps. In our 12-layer transformer, earlier layers (e.g., Layer 4, the Student) contain less refined representations, while deeper layers (e.g., Layer 8, the Teacher) produce more refined features. Both are extracted from the same forward pass at the same timestep $t$. The Teacher network uses Exponential Moving Average (EMA) weights to provide stable supervision to the Student’s shallower layer, creating a depth-wise consistency objective. We acknowledge the '$\Delta t$' notation in the text may have been ambiguous and will remove it in the revision to ensure clarity.
> Regarding the concern that SERE lacks rigorous analysis, we respectfully point to the extensive evaluations already included in the manuscript:
> The row 'Pert2Mol-wo-SERE' in Table 1 directly ablates the SERE component, showing it is critical for performance (e.g., improving FCD from 6.809 to 4.996). Table 3 further isolates SERE's impact on retrieval tasks, demonstrating a rise in MRR from 0.7656 to 0.8364.
> Section 3 provides the complete formulation, including the EMA update rule, specific layer selection, and projection head architecture.
> The loss curves explicitly visualize the method's stabilization effect, reducing final validation loss from 0.43 (without SERE) to 0.10 (with SERE).
> We believe these results constitute a rigorous empirical validation of the method's contribution."
>
> # Final comment to reviewer EBao
> **Pert2Mol stands alone as the first framework to integrate both transcriptomic and morphological data for molecule generation.** While recent works like [1] and [2] have pioneered generation from transcriptomics or images respectively, no prior method integrates these complementary modalities. Furthermore, our bidirectional cross-attention explicitly models the dynamic shift between control and treatment states, a critical advance over static profile generation.
>
> While inspired by established EMA frameworks, SERE introduces a specific innovation for rectified flow: depth-wise consistency. Instead of comparing timesteps or views, SERE stabilizes the high-dimensional training landscape by aligning representations across network depths (Layer 4 vs. Layer 8). This architectural choice is validated by significant metric gains (FCD: 6.8 $\to$ 5.0; MRR: 0.77 $\to$ 0.84) and a 4x reduction in final validation loss. We respectfully point to the empirical evidence, specifically the ablations in Tables 1 & 3 and the convergence analysis demonstrates rigorous validation.
>
> **We emphasize again, that we have presented the first benchmark for multi-modal phenotypic drug discovery, a novel flow-matching architecture, and a proven stabilization mechanism.** We have carefully curated the first multi-modal perturbation dataset and preprocessed from large number of samples across 3 different Ginkgo Bioworks' GDPx open datasets, to set a benchmark for future work to build on. We will release the full preprocessed datasets on Zenodo along with our code and training checkpoints for all models and baselines along with ablations.
>
> We hope these clarifications highlight the significant value Pert2Mol adds to the field.
>
> Reference:
>
> [1] Ohlan, Rahul, et al. "MolGene-E: Inverse Molecular Design to Modulate Single Cell Transcriptomics." bioRxiv (2025).
>
> [2]Zapata, Paula A. Marin, et al. "Cell morphology-guided de novo hit design by conditioning GANs on phenotypic image features." Digital discovery 2.1 (2023): 91-102.

---

### Comment · Area_Chair_Je3m · 2025-11-27

Thank you very much for the reviewer's comments and the author's positive response. As there is not much time left for discussion, please actively participate in the discussion and provide a more valuable response to this paper.

---

> ### Author Response · Authors · 2025-11-29
> **Thanking AC for support**
>
> Dear Area Chair,
>
> Thank you so much for your support! We are committed on improving the paper, and adding any experiments or clarifications that the reviewers and AC request for the final version. We believe our reviewers' area of expertise did not match the targeted research area of our submission (applications to physical sciences (physics, chemistry, biology, etc.)).
>
> We just want to re-iterate the main objective of the paper, and the proposed advancement we bring to multi-modal research in drug molecule design to be clear.
>
> **We emphasize, that we have presented the first benchmark for multi-modal phenotypic drug discovery, a novel latent flow-matching method with a new architecture, and a proven self representation based stabilization mechanism.** We have carefully curated the first multi-modal perturbation dataset and preprocessed from large number of samples across 3 different Ginkgo Bioworks' GDPx open datasets, to set a benchmark for future work to build on. We will release the full preprocessed datasets on Zenodo along with our code and training checkpoints for all models and baselines along with ablations.
>
> We are not simply showing an incremental upgrade in performance and numbers on an existing single modality task. **We are presenting and defining a completely new task of multi-modal phenotypic drug discovery. We curate a new dataset, create a new model architecture with state of the art generative methods, create a new benchmark on 3 tasks for future work to build on.** We believe this is a significant advancement in this area of research, and we hope the reviewers and AC understand the scope of our work.
>
> We hope these clarifications highlight the significant value Pert2Mol adds to the field.
>
> Regards,
>
> The Authors

---

### Note · Authors · 2026-01-26

I have read and agree with the venue's withdrawal policy on behalf of myself and my co-authors.

---

### Meta-Review · Area_Chair_K1mM · 2026-01-05

**Summary:**

After carefully considering the three reviews and the authors' responses, this paper proposes a valuable framework for a novel task—multi-modal phenotype-driven drug molecule generation—and presents preliminary experimental results. However, it does not meet the acceptance criteria for ICLR, primarily for the following reasons:

1. Limited Methodological Novelty: The core architecture is largely a composed integration of established components (ResNet, Transformer, rectified flow).

2. Insufficient Experimental Rigor: Despite the authors' defense highlighting their multi-modal ablation study, the evaluation lacks a systematic and quantitative comparison against state-of-the-art single-modal methods (e.g., transcriptomics-only generators).

3. Defensive Response and Unaddressed Concerns: The authors' replies, in part, appear defensive rather than constructively addressing the core weaknesses identified.

4. Task Definition vs. Technical Contribution Mismatch: While the paper defines a new multi-modal task and contributes a newly curated dataset, the methodological advancement and experimental depth do not yet rise to the level expected for ICLR.

In summary, I recommend rejecting the current version. The authors are encouraged to deepen the analysis of their method's novelty, expand the scope of experimental comparisons, provide a more rigorous theoretical and empirical grounding for components like SERE, and consider resubmission after substantial revision.

**Reviewer Concerns:**

Insufficient Experimental Rigor: Despite the authors' defense highlighting their multi-modal ablation study, the evaluation lacks a systematic and quantitative comparison against state-of-the-art single-modal methods (e.g., transcriptomics-only generators). As raised by Reviewer #3, the omission of benchmarking against relevant prior work weakens the claim of substantial performance improvement. The comparison primarily against a single diffusion baseline, while internally controlled, does not adequately situate the work's advantage within the broader field.

Defensive Response and Unaddressed Concerns: The authors' replies, in part, appear defensive rather than constructively addressing the core weaknesses identified. For instance, in the absence of direct multi-modal baselines, a more compelling evaluation strategy was not proposed. Key methodological clarifications requested by Reviewers #1 and #2—regarding the molecular encoder's specifics and the precise mechanism and novelty of SERE—were not fully resolved, leaving lingering doubts about the paper's technical contributions.

Task Definition vs. Technical Contribution Mismatch: While the paper defines a new multi-modal task and contributes a newly curated dataset, the methodological advancement and experimental depth do not yet rise to the level expected for ICLR. The conference typically seeks work with significant breakthroughs in methodological innovation, theoretical insight, or experimental thoroughness, areas where this submission has not fully demonstrated its competitiveness.

**Reviewer Scores:**

All three reviewers consistently scored the submission 2 (Reject), and their final assessment seems to be remained unchanged after the author's rebuttal.

Reviewer Ebao seems to maintain the score, as the authors' defensive reply did not adequately address the core weakness: insufficient justification and ablation for the novel SERE component.

Reviewer FH2E seems to maintain the score. While the novelty of the multi-modal task was clarified, the reviewer's fundamental concern about limited methodological innovation persisted.

Reviewer UD76 seems to maintain the score. The authors correctly noted a misunderstanding about multi-modal baselines, but the critical issue—the lack of comparison against advanced single-modal state-of-the-art methods—remained unresolved.

The rebuttal did not alleviate the major concerns regarding novelty or experimental rigor.

---

### Decision · Program_Chairs · 2026-01-26

Reject